

# AI-driven balance evaluation: a comparative study between blind and non-blind individuals using the mini-BESTest

Milagros Jaén-Vargas[1,2], Josué Pagán[3,4], Shiyang Li[1], María Fernanda Trujillo-Guerrero[1], Niloufar Kazemi[1], Alessio Sansò[1], Benito Codina-Casals[5,6], Roy Abi Zeid Daou[7] and Jose Javier Serrano Olmedo[1,8]

[1] Bioinstrumentation and Nanomedicine Laboratory, Center for Biomedical Technology (CTB), Universidad Politécnica de Madrid, Madrid, Spain
[2] Instituto Nacional de Investigaciones Científicas Avanzadas en Tecnologías de Información y Comunicación (INDICATIC AIP), Panama City, Panama
[3] Department of Electronic Engineering, Universidad Politécnica de Madrid, Madrid, Spain
[4] Center for Computational Simulation, Universidad Politécnica de Madrid, Madrid, Spain
[5] Didactic and Educational Research Department, Universidad de La Laguna, San Cristóbal de La Laguna, Spain
[6] Spanish Blind Organization (ONCE), Santa Cruz de Tenerife, Spain
[7] Faculty of Engineering–Polytech, Biomedical Engineering Department, Université La Sagesse, Furn El Chebbak, Lebanon
[8] CIBER-BBN, Centro de Investigación Biomédica en Red en Bioingeniería, Biomateriales y Nanomedicina, Madrid, Spain

Corresponding author
Jose Javier Serrano Olmedo, josejavier.serrano@upm.es

## ABSTRACT

There are 2.2 billion visually impaired individuals and 285 million blind people worldwide. The vestibular system plays a fundamental role in the balance of a person related to sight and hearing, and thus blind people require physical therapy to improve their balance. Several clinical tests have been developed to evaluate balance, such as the mini-BESTest. This test has been used to evaluate the balance of people with neurological diseases, but there have been no studies that evaluate the balance of blind individuals before. Furthermore, despite the scoring of these tests being not subjective, the performance of some activities are subject to the physiotherapist's bias. Tele-rehabilitation is a growing field that aims to provide physical therapy to people with disabilities. Among the technologies used in tele-rehabilitation are inertial measurement units that can be used to monitor the balance of individuals. The amount of data collected by these devices is large and the use of deep learning models can help in analyzing these data. Therefore, the objective of this study is to analyze for the first time the balance of blind individuals using the mini-BESTest and inertial measurement units and to identify the activities that best differentiate between blind and sighted individuals. We use the OpenSense RT monitoring device to collect data from the inertial measurement unit, and we develop machine learning and deep learning models to predict the score of the most relevant mini-BESTest activities. In this study 29 blind and sighted individuals participated. The one-legged stance is the activity that best differentiates between blind and sighted individuals. An analysis on the acceleration data suggests that the evaluation of physiotherapists is not completely adjusted to the test criterion. Cluster analysis suggests that inertial data are not able to distinguish between three levels of evaluation. However, the performance of our models shows an F1-score of 85.6% in predicting the score evaluated by the mini-BESTest in a binary classification problem. The results of this

study can help physiotherapists have a more objective evaluation of the balance of their patients and to develop tele-rehabilitation systems for blind individuals.

## INTRODUCTION

Vision is a crucial factor in the development of postural reflexes (*Alotaibi et al., 2016*). Blind individuals face postural issues due to the disruption of neurological processes caused by the loss of approximately half of the sensory input. Vision is essential for maintaining balance (*Stones & Kozma, 1987*; *Tomomitsu et al., 2013*), as it provides information about the environment and the body's position in space. While tactile and auditory senses are enhanced, they are less effective in maintaining proper biomechanics. To maintain balance, blind individuals adopt a gait with head retraction, increased pelvic rotation, excessive backward trunk lean, compensating forward head posture, abnormal trunk and arm movements, and flexion contractures, leading to faulty body mechanics as a natural compensation process.

Research has explored posture in visually impaired people (VIP). *Alghadir, Alotaibi & Iqbal (2019)* investigated postural stability by comparing the velocity of the center of gravity (COG) between blind and sighted individuals, showing that the COG velocity of subjects with visual impairment behaves similarly to that of sighted subjects with closed eyes. Furthermore, the mean COG velocity was higher in the visually impaired group than in the sighted group with open eyes, indicating that the visually impaired group may have a higher risk of falling than the sighted group.

Balance disturbances pose a significant challenge for people with visual impairments, which significantly affects their mobility, independence, and overall quality of life (*Jeon & Cha, 2013*). According to the World Health Organization, there are 2.2 billion visually impaired individuals worldwide. Of these, 285 million are blind, while the remaining 2 billion have low vision (*World Health Organization, 2023*). In the US only, in 2016, 36.8 million people reported a balance problem (*Mitchell & Bhattacharyya, 2023*). To address this issue, physical therapy interventions, often incorporating balance exercises, are commonly prescribed (*Mohammadkhani et al., 2021*). While these interventions have shown promise in improving balance in this population (*Zarei & Norasteh, 2022*), the effectiveness of these exercises can be significantly enhanced through regular monitoring and feedback.

Traditionally, balance assessments are conducted in clinical settings, limiting the frequency of evaluations and hindering the ability to track progress over time. To bridge this gap, there is a critical need for accessible and objective balance measurement tools suitable for home use (*Kelly et al., 2021*). Such tools would empower people with visual impairments to actively participate in their rehabilitation process by independently

monitoring their balance improvements and allowing healthcare professionals to provide timely feedback and adjustments to exercise regimens (*Rutkowska et al., 2015*).

Furthermore, recent studies have shown the potential of home-based exercise programs to improve balance in people with visual impairments. *Omidi et al. (2019)* highlighted the effectiveness of home exercises in enhancing balance, while *Yang et al. (2016)* explored the feasibility and potential benefits of home-based virtual reality balance training for individuals with motor impairments, including Parkinson's disease. In addition, *Haibach-Beach, McNamera & Lieberman (2022)* investigated the effectiveness of a home-based balance exercise program for older adults with visual impairments. Building upon these findings, the development of a user-friendly and reliable home-based balance assessment tool could contribute significantly to the management and rehabilitation of balance disorders in the visually impaired population.

Balance assessment is a cornerstone in the management of individuals with visual impairments. Several standardized tests have been developed to evaluate balance, each with its own strengths and limitations. The Berg Balance Scale (BBS) is a 15–20 min commonly used measure of functional balance, particularly in the elderly population (*Miranda-Cantellops & Tiu, 2022*). The Balance Evaluation Systems Test (BESTest) provides a comprehensive assessment of postural control systems that is suitable for patients with Parkinson's disease and the elderly (*Horak, Wrisley & Frank, 2009*). It takes about 35 min to be completed (*Franchignoni et al., 2010*). The mini-BESTest is the shorter version of the BESTest focusing on dynamic balance (*Di Carlo et al., 2016*). It takes 10–15 min to complete and is suitable for assessing balance in community-dwelling adults, elderly individuals, and patients with Parkinson's disease and stroke. The Y Balance Test (Y-Test) is a dynamic balance assessment used in sports medicine (*Gil-Martín et al., 2021*) and commonly utilized in clinical practice and research (*Johnston et al., 2016*).

While these tests offer valuable insights into balance function, they often require specialized training to administer and interpret them, limiting their accessibility in real world settings. Moreover, the time-consuming nature of these assessments can hinder their routine implementation in clinical practice (*Wei, McElroy & Dey, 2020*). Therefore, there is a growing need for user-friendly and objective balance measurement tools that can be easily administered in various environments, including home settings.

Previous research has explored the potential of simpler balance assessments, such as the Dynamic One-Leg Stance (DOLS), for individuals with visual impairments (*Blomqvist & Rehn, 2007*). The mini-BESTest have shown potential to be used in the evaluation of balance in people suffering from chronic stroke (*Tsang et al., 2013*), subacute stroke (*Inoue et al., 2024*), Parkinson's disease (*Franchignoni et al., 2022*; *Lopes et al., 2020*), or other neurological diseases (*Caronni et al., 2023*). However, the optimal balance assessment protocol for this population remains elusive. *Daneshmandi, Norasteh & Zarei (2021)* emphasized the need for qualitative studies to identify the most efficient balance assessment approach for individuals with visual impairments.

Traditional balance assessment methods, such as force plate platforms, while providing accurate measurements, are often expensive and lack portability, hindering their application in home-based settings (*Chen et al., 2021*). To overcome these limitations,

there is growing interest in exploring alternative approaches that leverage emerging technologies. IMUs are incorporated into wearable devices to gather data on human movement. The integration of artificial intelligence (AI) and wearable devices, specifically inertial measurement units (IMUs), offers promising avenues for developing innovative solutions to address the challenges associated with balance assessment in individuals with visual impairments.

While camera-based systems have been employed for balance analysis, their invasiveness and reliance on 2D estimations limit their effectiveness (*Milosevic, Leardini & Farella, 2020*). In contrast, IMUs provide three-dimensional real-time motion data, enabling accurate and objective balance assessments (*Noamani et al., 2020*).

Previous research has demonstrated the potential of IMUs and AI techniques for balance evaluation (*Bao et al., 2019*; *Kamran et al., 2021*; *Lin et al., 2022*). *Kim et al. (2021)* used a one-dimensional (1D) convolutional neural network (CNN) and gated recurrent unit (GRU) ensemble model to assess clinical balance using BBS. They compared it with previous work that used machine learning (ML) instead of deep learning (DL), obtaining 98.4% as the best accuracy of the model. Also, *El Marhraoui et al. (2023)* utilized a CNN to train and extracted the weights after training to predict fall events using the Balance Test Score.

These technologies have been successfully applied in various clinical settings, for example, in populations with neurological impairments, such as stroke (*Kim et al., 2021*). *Wei, McElroy & Dey (2020)* showcased the feasibility of combining IMUs, cameras, and Deep Learning for on-demand balance assessment. However, the application of IMUs for balance assessment in individuals with visual impairments, particularly in the context of home-based tele-rehabilitation, remains largely unexplored.

A comprehensive review of the literature revealed a lack of research investigating the application of IMUs to assess balance in individuals with visual impairments, particularly in the context of identifying optimal exercise protocols within the mini-BESTest framework. Although the potential of IMUs for the evaluation of balance in other populations, such as individuals with Parkinson's disease, has been explored (*Silva-Batista et al., 2023*), their application to the visually impaired population remains a largely unexplored area.

Traditional balance tests often rely on subjective evaluation, which can introduce bias and variability in the results. The use of objective measurements, such as those obtained from IMUs, can mitigate these issues by providing consistent and quantifiable data. This approach not only enhances the accuracy of balance assessments, but also facilitates the development of personalized rehabilitation programs.

The objectives of this study are: (i) to analyze whether the level of balance using the mini-BESTest score differs between blind and non-blind individuals, and to identify the most distinct exercise within the test in order to prioritize its improvement by professionals, thus avoiding the necessity of executing the entire test. (ii) Investigate the feasibility of employing AI techniques to automatically classify the level of performance achieved in the most significant exercises based on IMUs. (iii) Finally, to explore potential

alternative groups of participants and scores, unsupervised clustering techniques will be employed.

By focusing on specific balance tasks derived from the mini-BESTest, we seek to identify the most relevant exercises for monitoring balance progression in the blind population. This research has the potential to inform the development of a user-friendly, home-based balance assessment tool that can facilitate tele-rehabilitation and improve outcomes for individuals with visual impairments. In addition, this work contributes to the improvement of balance assessment practices and ultimately enhances the quality of life of this population.

## METHODOLOGY

### Experimental setup

To assess balance in blind and sighted individuals, OpenSense RT was used. OpenSense RT is a prototype developed by *Slade et al. (2021)* at Stanford University. The system was built around a Raspberry Pi 4 that controls a set of IMU sensors. This study used 12 IMU strategically placed on various body segments: pelvis, left and right feet, left and right shanks, left and right hands, left and right arms, torso, C5, and head. Figure 1 shows the placement of the IMUs in the body.

### Data acquisition

The IMUs used in this study were mounted on an AdaFruit ISM330DLC board, which is a six-axis IMU equipped with an accelerometer and a gyroscope (*Adafruit, 2024*). All sensors were wired to an I2C mux connected to a Raspberry Pi 4, which was responsible for data acquisition. The data were stored locally on an SD card for subsequent analysis. The Raspberry Pi 4 was powered by a battery and both components were placed in the pelvis, with all elements attached to the body using Velcro. The data acquisition process was initiated and terminated using an I2C button. The OpenSense RT system was configured to operate at a sampling frequency of 50 Hz, thereby capturing triaxial acceleration and gyroscopic measurements.

### Experimental conditions

The data acquisition took place in a controlled indoor environment with ambient lighting and minimal disturbances. Participants were asked to perform static and dynamic balance tasks. Prior to the session, they received instructions to maintain natural movements throughout the experiment. Participants were instructed to press a button to start, wait for the voice signal to begin, perform the task at a steady pace, and remain still once finished. They were then instructed to press the button again to end the task and confirm that the data had been recorded. The same instructions were given to blind and sighted people to ensure consistency.

### Subjects

To estimate the number of participants required to have a statistical power of 0.8, references are made to the assumptions of *Rutkowska et al. (2015)*. Their study

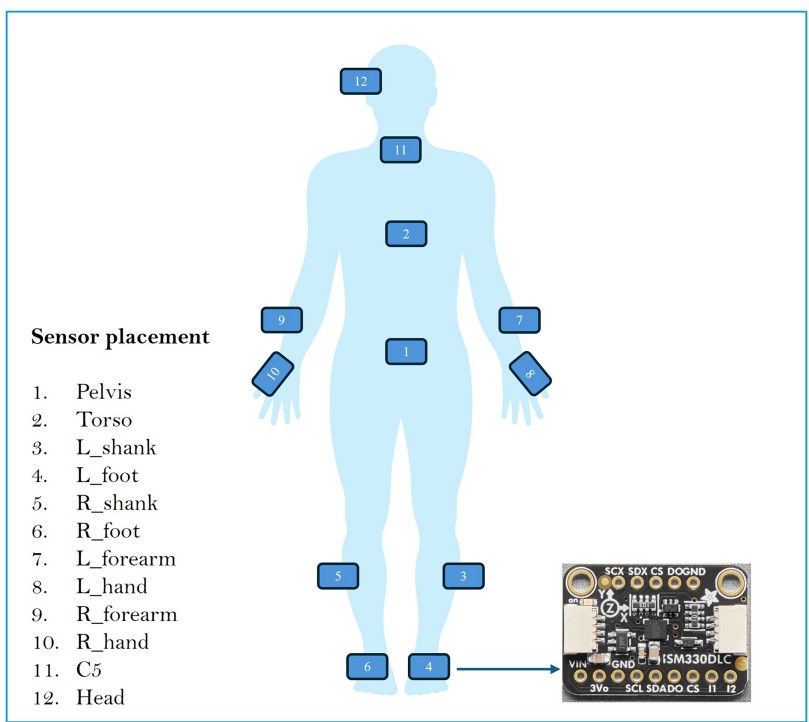

**Figure 1 Sensor placement in the body.**

demonstrated that at least 75% of the sighted individuals had a balance level above the average, while only 22% of blind individuals achieved a similar balance level. These percentages were used as reference values for our calculations to determine the necessary sample size. Setting an alpha value of 0.05, the resulting minimum number of participants equally divided between sighted and blind volunteers was 26. A total of twenty-nine volunteers were recruited for this study: fifteen sighted individuals and fourteen blind individuals. Table 1 shows the details of the sociodemographic variables of the participants in this study. The number of participants is higher compared to some in the state-of-the-art *Reyes Leiva, Gato & Olmedo (2023)*, *Nair et al. (2022)*, *Stearns et al. (2018)* with nine, 11 and 12 participants, respectively. The study was conducted in two locations: Centro de Tecnología Biomédica (CTB) and Organización Nacional de Ciegos de España (ONCE), Madrid, Spain.

The study objectives, experimental protocol, and informed consent were approved by the Ethics Committee of the Universidad Politécnica de Madrid (UPM) (ADPCEPCDVY-JJSO-DATOS-20230329).

## Expert evaluation of mini-BESTest

A physiotherapist and a blind mobility teacher adapted the mini-BESTest for use with blind individuals by reducing the duration of the exercises, thus allowing blind participants to complete them. To illustrate this point, in Test 3, the maximum duration was reduced from 15 to 10 s. The test was adapted from the Spanish translation performed by

**Table 1 Mean values and standard deviation of values of sociodemographic variables.**

| Cohort | Male/Female (%) | Age (years) | Height (m) | Weight (kg) | Practice sport (%) | Location |
|---|---|---|---|---|---|---|
| Sighted | 6/9 (40.0/60.0) | $25.9 \pm 5.5$ | $1.65 \pm 0.07$ | $63.5 \pm 8.1$ | 14 (93.3) | CTB |
| Blind | 6/8 (42.9/57.1) | $40.1 \pm 10.6$ | $1.65 \pm 0.09$ | $69.9 \pm 10.2$ | 11 (78.6) | ONCE |
| Total | 12/17 (41.4/58.6) | $32.8 \pm 10.9$ | $1.65 \pm 0.08$ | $66.6 \pm 9.6$ | 25 (86.2) | |

Dominguez-Olivan et al. in *Franchignoni et al. (2017)*. Each participant completed the mini-BESTest in approximately 1 h.

The activities were recorded on video for subsequent evaluation by the two experts. The exercises were assigned a rating of 0, 1, or 2, corresponding to bad, mild, and good balance, respectively. The total possible score for the exercises was 28 points. In instances where exercises were designed to be performed on both the left and right sides, the final evaluation considers the worst-case scenario.

Cohen's Kappa coefficient was calculated for the fourteen activities of the test. This coefficient quantifies the average concordance between the two raters. The results showed a highly satisfactory Cohen's Kappa index of 0.87. Thus, it was decided to use for the rest of the manuscript the punctuation determined by expert 1.

## Methodology on modeling and data analysis

A comprehensive methodology was developed to analyze the feasibility of using IMU data to classify the mini-BESTest. The methodology was divided into three main stages. The methodology workflow is illustrated in Fig. 2, which also highlights the main steps of the study as follows:

1. A statistical analysis of the mini-BESTest evaluation performed by the physicians, followed by and compared to a statistical analysis of the duration of the IMU data of the most representative exercise.

2. A supervised classification strategy using information from the IMU data: (i) based on signal similarities using Dynamic Time Warping (DTW) and classification with K-Nearest Neighbors (KNN). DTW was chosen for its ability to align and compare time series with temporal distortions, while KNN was selected for its simplicity, robustness, and effectiveness in clustering, particularly compared to other methods such as DBSCAN; (ii) based on time features extracted from the signals and classification with Random Forest, a robust and versatile algorithm known for its ease of implementation and generally strong performance across various datasets; and (iii) based on the raw signals and classification with advance Deep Learning models, such as CNN + LSTM, to leverage their capability to capture complex temporal patterns and explore modern approaches for time series analysis.

3. Finally, two cluster analyses were performed to identify whether there were subgroups of participants or scores based on (i) sociodemographic information and mini-BESTest scores, and (ii) IMU data, respectively.

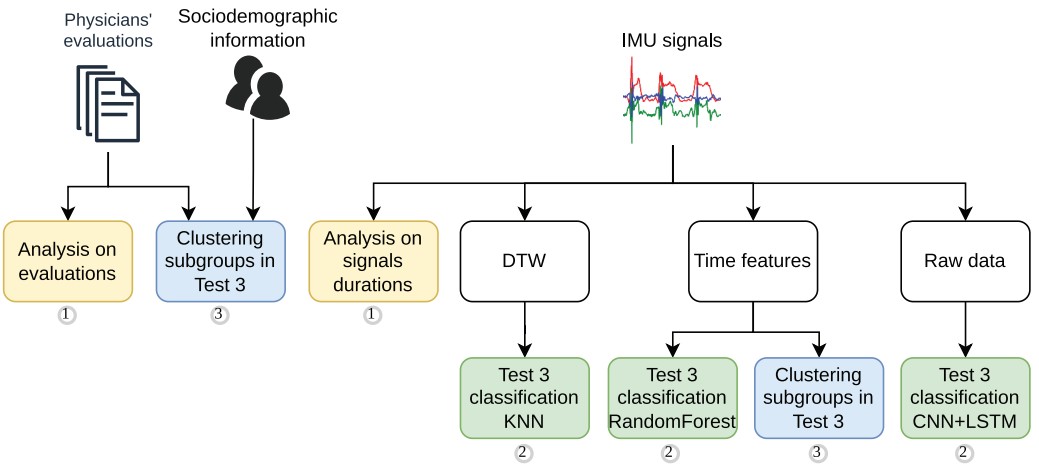

**Figure 2** Methodology workflow for the signal processing, feature extraction, data analysis and supervised modeling.

## Statistical analysis on mini-BESTest evaluation and signal duration

Several statistical analysis tests were performed during the study. The two main analyses are shown in Fig. 2. To analyze the differences in the mini-BESTest evaluations between blind and sighted individuals, an independent sample t-test was used. This process was applied in the analysis during the duration of the IMU signals. The significance level was set at $p = 0.05$.

For the feature selection carried out in the cluster analysis using sociodemographic information and mini-BESTest scores, the Recursive Feature Elimination (RFE) algorithm using a Pearson correlation coefficient analysis was carried out. Pearson's correlation coefficient was used to analyze the relationship between the mini-BESTest scores and the sociodemographic information of the participants. To do so, the `scikit-learn` Python library was utilized.

## AI-driven classification methods

Before any classification task, inertial data were processed as follows: 1 s was removed at the beginning and end of the signals to avoid using the data when the participant was not performing the activity.

In all studies, a 5-CV strategy was employed to evaluate the models with stratified subject-wise splits. The imbalance in tabular data collected from IMUs (DTW and time features) was corrected in the training data using the Synthetic Minority Over-sampling Technique (SMOTE) algorithm (`imbalanced-learn` Python library v0.12.3). The models were trained for multiclass and binary classification grouping either bad and mild evaluations or mild and good evaluations. The performance metrics used were accuracy, precision, recall and the F1-score weighted on the test folds. The models were implemented and evaluated using the `scikit-learn` and `keras` Python libraries.

In order to have a larger dataset, activities involving the evaluation of both legs were considered separately.

We conducted an exploratory analysis to determine the minimum number of sensors required for an accurate classification. The analysis was based on the importance of the sensors in the classification of the mini-BESTest guided by experts and also based on the body parts that were less relevant to movement. Consequently, up to six sensors were excluded from the analysis in different experiments, including the head, neck (C5), left and right forearms, and left and right feet.

### DTW and KNN classification

As previously stated, the first classification task was based on the similarity of the signals using DTW and the classification with KNN. The DTW algorithm was computed by comparing each axis (X, Y, Z) of each of the 12 sensors between all the participants. Thus, the matrix of distances evaluated had $3 * 12 * \binom{29}{2} = 14{,}616$ comparisons on each leg. It was used the `fastdtw` Python library to compute the DTW distance. The KNN algorithm was used to classify the participants according to the DTW distance.

Following an exploratory analysis, the sensors that were most relevant for the classification were selected. The sensors selected for this purpose included the pelvis, right foot, left foot, right hand, left hand, and head. The duration of the recording was also included as a feature. No other sociodemographic information was included. A sequential forward feature selection with a KNN estimator ($k = 3$) and the 3-CV method was used to select the most relevant features. The value $k = 3$ was chosen because it is the default parameter in the `sklearn` Python library and is appropriate given the limited number of participants, as larger values would dilute the classification performance. Feature subsets of 20% and 50% were tested to balance the reduction of dimensionality with the retention of information, with no significant differences in the results, demonstrating the robustness of the model. After feature selection, the DTW data were standardized. A KNN classification was performed with $k = [1 : 28]$ neighbors, covering the feasible range based on the number of participants. The classification was performed using the Euclidean distance.

### Time features and random forest classification

The second classification task was based on four time features extracted from the accelerometer signals and classification with Random Forest (refer to Fig. 2). We extracted 144 features including: mean, standard deviation, energy, and amplitude for each axis of the 12 sensors. The duration of the recording was also added as a feature. No other sociodemographic information was used. The features were computed using an overlap sliding window. Several combinations of window sizes $ws = [0.5, 1, 1.5]$ s and overlap times $ot = [0, 0.5]$ s were studied. Window sizes smaller than 0.5 s were avoided since the movements of interest are not rapid enough to justify shorter windows, while window sizes larger than 1.5 s were excluded because some activities are brief, and larger windows could reduce classification performance by averaging out relevant information. The overlap times were selected to align with the range of window sizes tested, ensuring sufficient data coverage while avoiding redundancy.

A sequential forward feature selection with the same parameters as in the DTW experiment was used to select the most relevant features. After that, the Random Forest algorithm was used to classify participants according to the selected time features. The Random Forest algorithm was implemented using the `sklearn` Python library. A grid search was performed to find the best hyperparameters for the Random Forest algorithm. The hyperparameters tuned were: number of estimators $ns = [25, 50, 500, 1,000]$, maximum depth $md = [10, 20, 30]$, minimum sample split $mss = [2, 5, 10]$, and minimum sample leaf $msl = [2, 5, 10]$. The range of values for the number of estimators $ns$ was chosen based on standard practices for datasets of this size, as these values balance computational efficiency and model performance.

The windowing led to a data set in which different participants had different numbers of samples. For training, each window was assigned the same label as the participant. In the testing phase, only one label per participant was used. To achieve this, the final prediction was selected as the most prevalent label for each participant.

### Deep learning classification

The third classification task was based on raw signals and classification with a DL model (Fig. 2). The DL models were fed alternatively with two kinds of raw signals: (i) the raw acceleration signals from the IMUs, and (ii) the quaternions obtained from the IMUs. A quaternion is a mathematical representation of a rotation in a three-dimensional space. The quaternions were obtained from the acceleration and gyroscope data using the Mahony filter to estimate the orientation of the sensor. The Attitude and Heading Reference Systems (AHRS) Python library was used to compute the quaternions.

Both raw signals were windowed and standardized using the Standard Scaler function from the `sklearn` Python library. The signals were segmented using a sliding window technique (*Reyes Leiva, Gato & Olmedo, 2023*). The window sizes used were $ws = [0.5, 1]$ s with fixed overlap time $ot = 20$ ms to do the data augmentation necessary for these kinds of models.

Four DL architectures were tested: CNN, LSTM, GRU, and two architecture variations of a CNN-LSTM hybrid model. Table 2 shows the configuration of the models' architectures. These network architectures were chosen at the discretion of the experimenter after a number of iterations of exploratory analysis. All models were trained with: `batch size` $= [32, 64]$, `epochs` $= [25, 50, 100]$, `learning rate` $= 0.001$, `optimizer` $= Adam$, $ReLU$ as `activation function`, and `loss=` *categorical crossentropy*. The models were implemented using the `keras` Python library.

It is worth mentioning that in the exploratory analysis time-series data augmentation was employed to correct the imbalance of the dataset. In order to achieve this objective, the raw signals were augmented with different techniques, such as time change, time warping, and amplitude modulation, using the `tsaug` Python library. However, the results were not improved and the data augmentation was discarded. A total of 20% of the participants were used for validation and 20% for testing. Due to the reduction in the dataset resulting from the validation split, in this experiment a double nested 5-CV strategy was used. The models were trained using the `keras` Python library.

**Table 2 Architectures and hyperparameters tested on the evaluation of deep learning classification models.**

| Parameters | Layers information | Dropout rate |
|---|---|---|
| **CNN** | 1D-Conv (32, filter = 3, kernel = 0) → MaxPooling (2) → Flatten → Dense (64) → SoftMax | – |
| **LSTM** | LSTM (100) → Dense (100) → SoftMax | 0.5 |
| **GRU** | GRU (64) → GRU (32) → Dense (64) → SoftMax | – |
| **CNN-LSTM arch. 1** | 1D-Conv (filter = 16, kernel = 5) → 2 * 1D-Conv (filter = 64, kernel = 3) → MaxPooling (2) → Flatten, LSTM (20) → Flatten → Dense (20) → SoftMax | 0.5 |
| **CNN-LSTM arch. 2** | 1D-Conv (filter = 64, kernel = 3) → Flatten → LSTM (50) → Flatten → SoftMax | 0.5 |

## Machine learning clustering methods

In view of the results obtained and recognizing the difficulty in classifying the evaluations of the participants, a cluster analysis was performed to determine whether subgroups of participants could be identified. Specifically, two cluster analyses were conducted: (i) one based on sociodemographic information and mini-BESTest scores and (ii) another based on the features of the IMUs.

Both cluster analyses were performed using the K-means algorithm. The number of clusters $k$ ranged from $k = 1$ to $k = 9$, as we did not anticipate the presence of more than nine meaningful subgroups in the data.

To compare the performance of the cluster analysis, three metrics were used: the Within Cluster Sum of Squares (WCSS), the Silhouette score, and the Davies-Bouldin. The elbow method was used to determine the optimal number of clusters according to WCSS. To find the optimal number of clusters based on the Silhouette coefficient, we must select the number of clusters that maximize it. In contrast, to do so based on the Davies-Bouldin index, we must select the number of clusters that minimize it.

### Clustering based on sociodemographic information and mini-BESTest scores

The first cluster analysis was based on sociodemographic information and the mini-BESTest scores. A total of 23 features were used as input data: (i) the 14 individual punctuations and four aggregate punctuations of the mini-BESTest, and (ii) sociodemographic information on sex, age, height, weight, and sport practice. Before the cluster analysis, a feature selection was conducted by eliminating first high correlated features —those with absolute correlation higher than 0.9—, and then an RFE with linear regression approach was used. The features were standardized before the cluster analysis.

### Clustering based on IMU data

The second cluster analysis was based on time features. On the 144 features extracted from the IMU data, a feature selection was carried out, as in the previous cluster analysis, by eliminating first the high correlated features and then using RFE. The features were also standardized.

# RESULTS

This section presents the results of the study. The results are divided into four subsections following the division of experiments presented in the methodology section (see Fig. 2). The subsections are as follows: (i) evaluation of the mini-BESTest, (ii) duration analysis, (iii) automatic classification of the mini-BESTest score based on IMU data, and (iv) cluster analysis to identify if subgroups of participants exist.

## Statistical analysis on mini-BESTest evaluation and signal duration
### Expert evaluation of Mini-BESTest

In this section, it was analyzed whether mini-BESTest is capable of identifying differences in balance between blind and sighted individuals. Figures 3A and 3B depicts the punctuation histograms for each of the fourteen activities of the mini-BESTest for blind and sighted individuals.

As can be seen in Fig. 3, the punctuation of some activities was unable to identify differences between blind and sighted individuals, but some of them did. To better understand the differences in balance between blind and sighted individuals, a statistical analysis was performed.

The results shown in Table 3 suggest that there were no significant differences between blind and sighted individuals in the reactive postural control and sensorial orientation categories. However, there were significant differences in the categories of anticipatory postural control and dynamic gait, which is in line with the results of *Wiszomirska et al. (2013)* for visually impaired women. The overall mBESTest score was also significantly different between blind and sighted individuals, but mainly due to differences in the categories of anticipatory postural control and dynamic gait.

The average mini-BESTest score for all participants was $25.9 \pm 2.3$, leaving 86.7% of sighted individuals and 28.6% of blind individuals above the average. These results match the assumptions made in the sample size calculation. With these results, we can also say that the average balance of young sighted adults is $27.3 \pm 1.2$, while for blind individuals it is $24.7 \pm 2.1$. These insights can be used to compare the expected results with other cohorts of participants such as those described in *Almeida, Marques & Santos (2017)*.

To discern which of the activities in the anticipatory postural control and dynamic gait categories are responsible for the differences between blind and sighted individuals, a *post hoc* analysis was performed. The values are shown in Tables 4 and 5.

The evaluation of anticipatory postural controls involves the performance of tasks that necessitate deliberate and volitional movements that challenge the subject's ability to maintain equilibrium. Include Tests 1, 2, and 3, which correspond to the following tasks: sit to stand, rise to the toes, and stand on one leg. The mini-BESTest employs these tasks to assess an individual's capacity to anticipate and prepare for postural alterations.

Assessment of dynamic gait entails the performance of tasks that challenge the individual's capacity to maintain balance while walking under varying conditions. This assessment involves activities such as those tested in Tests 10, 11, 12, 13, and 14, including: changing speed, walking with head turns, walking with pivot turns, stepping over obstacles, and timed up and go with dual task. The mini-BESTest assesses the individual's capacity to

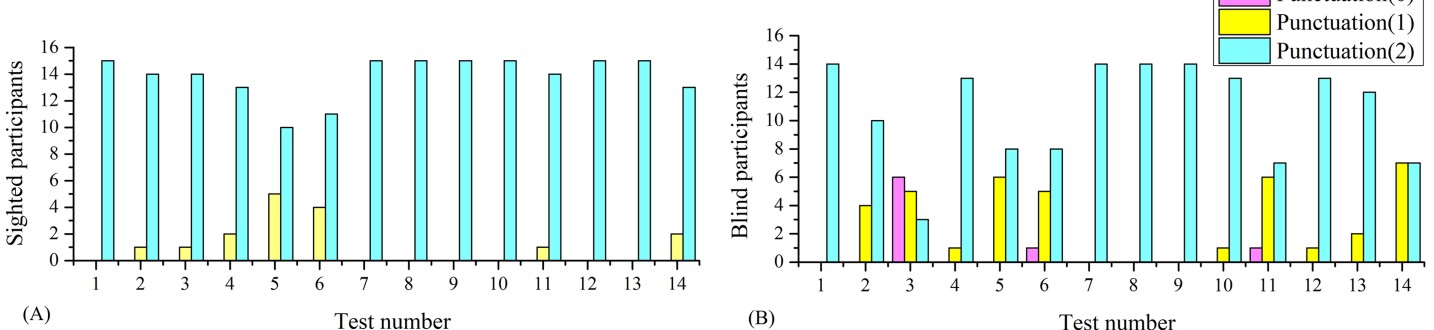

**Figure 3** **Histograms of the mini-BESTest for sighted and blind individuals.** (A) Histogram for sighted individuals. (B) Histogram for blind individuals.

**Table 3** **Average and standard deviation of partial and overall mini-BESTest punctuations for both blind and sighted.**

| | Anticipatory ($p = 0.0007$) | Reactive postural control ($p = 0.4045$) | Sensorial orientation ($p = 1.0$) | Dynamic gait ($p < 0.0001$) | Overall mBESTest score ($p = 0.0003$) |
|---|---|---|---|---|---|
| Sighted | 5.9 ± 0.4 | 5.4 ± 1.0 | 6.0 ± 0.0 | 10.0 ± 0.0 | 27.3 ± 1.2 |
| Blind | 4.6 ± 1.1 | 5.1 ± 0.9 | 6.0 ± 0.0 | 8.7 ± 1.3 | 24.4 ± 2.4 |

**Table 4** **Average and standard deviation of punctuations on the three tests of the anticipatory category for both blind and sighted.**

| | Test 1 ($p = 1.0$) | Test 2 ($p = 0.4508$) | Test 3 ($p < 0.0001$) |
|---|---|---|---|
| Sighted | 2.0 ± 0.0 | 1.9 ± 0.3 | 1.9 ± 0.3 |
| Blind | 2.0 ± 0.0 | 1.8 ± 0.4 | 0.8 ± 0.8 |

**Table 5** **Average and standard deviation of punctuations on the four tests of the dynamic gait category for both blind and sighted.**

| | Test 11 ($p = 0.0100$) | Test 12 ($p = 0.2071$) | Test 13 ($p = 0.3409$) | Test 14 ($p < 0.0001$) |
|---|---|---|---|---|
| Sighted | 2.0 ± 0.0 | 2.0 ± 0.0 | 2.0 ± 0.0 | 2.0 ± 0.0 |
| Blind | 1.5 ± 0.7 | 1.9 ± 0.3 | 1.9 ± 0.4 | 1.4 ± 0.5 |

manage these dynamic challenges, which are essential for safe and effective ambulation in daily life.

The results of the *post hoc* analysis in Tables 4 and 5 showed that the differences between blind and sighted individuals in the anticipatory postural control category were mainly due to Test 3. The differences in the dynamic gait category were mainly due to Test 11 and Test 14. The results suggest that the anticipatory postural control and dynamic gait categories were the most relevant to discern between blind and sighted individuals.

The results suggest that blind individuals have more difficulty maintaining balance while standing on one leg compared to sighted individuals. The total number of sighted individuals was evaluated with the highest value in Test 3, indicating a good balance. In contrast, blind individuals exhibited a spectrum of balance, including poor, mild and optimal balance: 28.6% of blind individuals had poor balance, 42.9% had mild balance, and 21.4% had good balance.

These results satisfied the objective (i) of this study. The mini-BESTest identified differences in balance between blind and sighted individuals. The differences were mainly due to the anticipatory postural control and dynamic gait categories, which were the most relevant to discerning between blind and sighted individuals.

Henceforth, the focus will be directed towards the Test 3 activity, as it is the most relevant to evaluate anticipatory capacity, where health professionals can concentrate their attention.

## Analysis of evaluations of Test 3 by signal duration

One of the objectives of this work is to analyze whether automatic evaluation of the activities of the mini-BESTest is possible. In particular, the focus will be on Test 3 (monopodal support) which, as presented earlier, is the activity with the greatest difference between blind and sighted individuals. According to the adapted evaluation test (*Franchignoni et al., 2017*) and the review of experts, to score the Test 3 activity, one must:

- Record the number of seconds that the subject can sustain, up to a maximum of 15 s, stopping the time when the subject moves his hands from his hips or puts one foot down.
- Include only the score for one side (the worst score) and select the best time of the two records (for one side) for the score.
- Award a score of 0 if the subject cannot stand on one foot for 5 s, 1 if they can stand between 5 and 10 s, and 2 if they can stand for more than 15 s.

Based on these rules, the first thing to think about is automating the score according to the duration of the exercise. As mentioned above, the duration of the IMU recording starts and stops when the participant presses the button, so to calculate the duration of the exercise, the time that elapses from when the participant presses the button until one foot is lifted off the ground, and from when the foot is placed on the ground until the button is pressed again, must be subtracted. Automating this is a complicated task, as in the same record there may be several attempts to perform the exercise, and therefore several exercise times. It has been decided to make a manual selection of the records by looking at accelerometer data, and subsequently, score the exercises. In this section, an analysis of the times and scores awarded by the evaluators for Test 3 was made.

Figure 4A shows the three axes of the right shank's accelerometer (standing on the left leg) of a well-performed exercise by a sighted participant. The record had a duration of 17.92 s, but the time between the acceleration peaks is 15.64 s. Figure 4B shows an example of a score of 0 rated by the evaluators for a blind participant standing on his right foot. In this case, the record duration is 9.66 s, but there are four attempts in the record. Due to

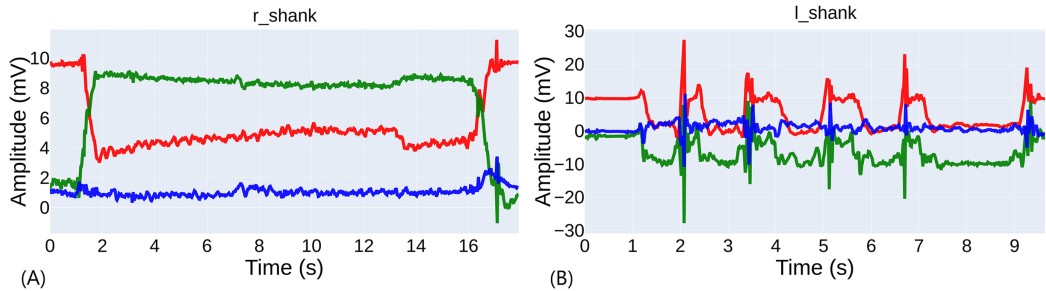

**Figure 4 Comparison of exercise performance.** (A) Example of a well-performed exercise. (B) Example of a poorly performed exercise with several attempts.

similar cases, it is complicated to automate the cutting of the records to consider only the last attempt. In this case, the time of the last attempt was 2.60 s. The average extra time in the records was $2.69 \pm 1.00$ s for the sighted and $3.27 \pm 1.06$ s for the blind. The difference between blind and sighted individuals was significant ($p = 0.0342$), so it can be inferred that blind individuals found more difficulty finishing the exercise, recovering the position and pressing the button.

The average duration of the exercises for both legs of the sighted participants was $14.49 \pm 4.00$ s, and $6.08 \pm 4.05$ s for the blind ($p < 0.0001$) with no significant difference between legs among the same group (refer to Table 6).

The average differences between the durations and the scores are significant when analyzing the entire dataset (see Table 7). The average difference between the durations of the exercises evaluated with 0 and 1 is 5.37 s ($p < 0.001$) and 5.75 s between the exercises evaluated with 1 or 2 ($p < 0.001$).

Referring to Table 7, it can be seen that there were only significant differences in the duration of the signals between blind and sighted individuals when scored with the maximum rating 2 ($p < 0.01$). The average duration of the blind for a score of 2 was $9.54 \pm 3.40$ s, and $14.51 \pm 4.07$ s for sighted. Sighted individuals lasted an average of 4.97 s longer to achieve a score of 2.

As can be inferred, there were many classifications of 2 in blind participants with a duration less than 10 s required by the evaluation sheet, which does not occur when the score was 1. In this case, the average duration of the blind was $6.71 \pm 2.83$ s; this was higher than the 5 s required by the evaluation sheet. The reason why evaluators have scored blind individuals with durations of less than 10 s with 2 is something that needs to be analyzed; it may be due to the difficulty in counting times by watching a video or to some subjective and positive bias toward the blind evaluation.

In search of these results, a re-evaluation of the exercises of the participants according to the manually cut activity durations was made. Tables 8 and 9 compared the results of the expert evaluation with the re-evaluation based on the manual inspection of IMU signals for sighted and blind participants, respectively. Most of the discrepancies occurred between punctuations 1 and 2. For example, two exercises were evaluated as 2 while their durations were 4.92 s (sighted) and 3.28 s (blind). The Cohen's Kappa coefficient between the evaluations of the experts and the re-evaluation is 0.596, indicating a fair-moderate

**Table 6** Average and standard deviation of duration on the performance of Test 3 for sighted and blind participants after manual cut of the IMU signals.

|  | Left leg (s) ($p < 0.0001$) | Right leg (s) ($p < 0.0001$) | Both legs (s) ($p < 0.0001$) | $n$ |
|---|---|---|---|---|
| **Sighted** | $14.01 \pm 2.73$ | $14.97 \pm 5.02$ | $14.49 \pm 4.00$ | 30 |
| **Blind** | $5.32 \pm 3.16$ | $6.84 \pm 4.78$ | $6.08 \pm 4.05$ | 28 |

**Table 7** Average and standard deviation of duration on the performance of Test 3 for sighted and blind participants for the different evaluation marks.

|  | 0 points (s) | 1 point (s) | 2 points (s) | $n$ of 0 | $n$ of 1 | $n$ of 2 |
|---|---|---|---|---|---|---|
| **Sighted + blind** | $2.12 \pm 0.60$ | $7.49 \pm 3.54$ | $13.24 \pm 4.45$ | 10 | 9 | 39 |
| **Sighted** | $0.00 \pm 0.00$ | $13.76 \pm 0.00$ | $14.51 \pm 4.07$ | 0 | 1 | 29 |
| **Blind** | $2.12 \pm 0.60$ | $6.71 \pm 2.83$ | $9.54 \pm 3.40$ | 10 | 8 | 10 |

**Table 8** Comparison of physicians evaluations and re-evaluation of Test 3 performance for blind people based on the manual inspection of IMU signals on both legs.

|  |  | Re-evaluation for sighted participants | | | |
|---|---|---|---|---|---|
|  |  | 0 | 1 | 2 | *Sum* |
| **Physicians evaluation** | **0** | 10 | 0 | 0 | 10 |
|  | **1** | 2 | 5 | 1 | 8 |
|  | **2** | 1 | 5 | 4 | 10 |
|  | *Sum* | 13 | 10 | 5 | 28 |

**Table 9** Comparison of physicians evaluations and re-evaluation of Test 3 performance for sighted people based on the manual inspection of IMU signals on both legs.

|  |  | Re-evaluation for sighted participants | | | |
|---|---|---|---|---|---|
|  |  | 0 | 1 | 2 | *Sum* |
| **Physicians evaluation** | **0** | 0 | 0 | 0 | 0 |
|  | **1** | 0 | 0 | 1 | 1 |
|  | **2** | 1 | 2 | 26 | 29 |
|  | *Sum* | 1 | 2 | 27 | 30 |

agreement between the evaluations. If separated between blind and sighted individuals, as shown in Tables 8 and 9 the agreement for the evaluation of the sighted was 86.7%, while for the blind was 67.9% with a Cohen's Kappa of 0.519. The agreement was higher in the sighted than in the blind, which may be due to the difficulty in evaluating the blind by experts and other errors in the evaluation process.

Using re-evaluations based on IMU signals to search for significant differences in Test 3, it can be seen that they were still significant, so Test 3 was still the most relevant. The

difference increased from 1.1 to 1.3 points between blind and sighted individuals ($p < 0.0001$).

In summary, all this analysis indicated that the evaluation of the exercises in Test 3 was complicated, and that the objective measurement of the duration of the exercises was an important factor to consider. The average duration of the exercises for blind individuals was lower than that of sighted individuals, and the agreement between the evaluations of the experts and the re-evaluation by time was moderate. This discrepancy had an impact on the modeling of the automatic classification of the score with supervised learning algorithms, and therefore this factor had to be taken into account in future work. However, the work continued with the evaluation of the experts, as it was considered the ground truth.

## AI-driven classification analysis

### DTW and KNN classification analysis in Test 3

The analysis of DTW distances between accelerometer signals of the different subjects aims to show if there is a significant difference between the signals of the subjects in general and according to the score in Test 3. In the statistical analysis of the DTW distance matrix, some interesting results stood out.

To perform a comparison of the distances between the signals of the subjects, the mean distance between the signals of the subjects was calculated. It is defined as the arithmetic mean of the distances of each acceleration axis of a subject to the rest of the subjects. The comparison was made by punctuation. The results are shown in Table 10.

The difference in means between the distances of the signals of subjects with a score of 1 and 2 was $331.73 \pm 95.96$, while the difference in means between the distances of the signals of subjects with a score of 2 was $367.87 \pm 82.17$. The comparison of these two distributions showed a significant difference with $p < 0.00001$. This means that there were differences between the signals of the subjects with a score of 1 and 2. In contrast, the fact that the remaining comparisons were not significant indicates that there were no significant differences between the signals of subjects with a score of 0 and 1 and between subjects with a score of 1 and 2, which may be symptomatic of the signals being more similar to each other and the classification being more complicated.

Going deeper into this comparison, and looking at Table 10, it can be asserted that it was in blind subjects where the distance between signals with a score of 1 and 2 ($288.45 \pm 85.05$) was lower than in sighted subjects ($427.28 \pm 101.16$). This again reveals that, in general, the signals of blind subjects with a score of 1 and 2 were more similar to each other than those of sighted subjects, which may also be a factor that affects the classification.

Other more detailed analysis suggested that the signals of only five subjects with vision (33.3%) had significant differences from other subjects and that seven blind subjects, who constituted 50% of the population, did not. This will also be a handicap for the classification.

As expected from the aforementioned complications, the classification of the signals of the subjects according to their score in Test 3 was tough. A KNN classification analysis was conducted and the best results are shown in Table 11. A weighted F1-score of $73.9 \pm 9.3$

**Table 10 DTW distances comparing the Test 3 score and cohort of participants.**

| Score 1st ind. | Score 2nd ind. | $n$ | DTW all participants | $n$ both blind | DTW blind | $n$ both sighted | DTW sighted |
|---|---|---|---|---|---|---|---|
| 0 | 0 | 21 | 268.46 ± 61.04 | 21 | 268.46 ± 61.04 | 0 | – |
| 0 | 1 | 44 | 296.47 ± 102.63 | 40 | 282.04 ± 59.98 | 0 | – |
| 0 | 2 | 194 | 329.81 ± 102.63 | 48 | 300.86 ± 76.04 | 0 | – |
| 1 | 1 | 16 | 303.57 ± 95.96 | 12 | 285.32 ± 59.28 | 0 | – |
| 1 | 2 | 176 | 331.73 ± 95.96 | 40 | 288.45 ± 85.05 | 14 | 427.28 ± 101.16 |
| 2 | 2 | 361 | 367.87 ± 82.17 | 21 | 332.44 ± 114.20 | 196 | 372.93 ± 116.36 |

was achieved for binary classification grouping bad and mild evaluations. This model exceeded the ground truth—which is 66.7—, however, it was not significantly different from the next best model for $k = 4$ and F1-score 68.9 ± 15.9 ($p = 0.4082$). The best result was obtained for a selection of 12 sensors: head, neck, torso, pelvis, left and right forearm, hands, shanks, and feet.

### Time features and random forest classification analysis in Test 3

Table 11 shows the best results of the classification analysis using time features and Random Forest. A weighted F1-score of 70.0 ± 13.6 was achieved for binary classification grouping bad and mild evaluations, which slightly exceeded the ground truth, which is 66.7. However, the difference was not significant ($p = 0.3069$) from the next best model with an F1-score of 62.0 ± 10.3. The best result was obtained for a selection of 24 features belonging to four sensors: the left and right shanks. It is also noteworthy that none of the selected features includes the energy of the signal.

As expected, the imbalance and errors in the evaluations lead to a decrease in the classification performance and lead to the application of a binary classification approach to improve the results. The result was lower than the DTW + KNN model, despite the fact that the difference is not significant ($p = 0.5535$).

### Deep learning classification analysis in Test 3

In this section, the classification analysis of Test 3 with Deep Learning and raw acceleration signal is shown. As mentioned in the methodology, different architectures, input data such as quaternions, and temporal windows was tested. The results of the best combination of architecture and input data are shown in Table 11.

The best result was obtained with a weighted F1-score of 85.6 ± 6.7 for binary classification grouping mild and good evaluations. This model exceeded the ground truth, which is 66.7, and the difference is significant ($p = 0.0272$) from the next-best model, which is DTW + KNN. However, the best result was obtained for the entire set of sensors. As a result, for the current results it was not possible to determine which sensors were most relevant for the classification of Test 3 and remove any of them.

Because the best model combines mild and good evaluations, in contrast to other models, it was evident that mild evaluations were the most difficult to classify. Nine out of thirteen incorrectly evaluated by duration were mild evaluations that should have been good or good evaluations that should have been mild.

**Table 11  5-CV results of experiments on classification of balance performance for Test 3.**

| Experiment | Accuracy weighted | Precision | Recall | F1 weighted | Best setup | Most important sensors |
|---|---|---|---|---|---|---|
| DTW + KNN | 72.1 ± 12.0 | 76.8 ± 12.3 | 74 ± 9.4 | 73.9 ± 9.3 | Binary classification (grouping: mild & good). $k = 5$ | 12 sensors: head, neck, torso, pelvis, left and right forearm, hand, shank and foot |
| Time features + RF | 64.4 ± 13.9 | 73.6 ± 18.6 | 75.7 ± 9.6 | 70.3 ± 13.6 | Binary classification (grouping: mild & good). `max depth = 20, min samples leaf = 5, n estimators = 50, random state = 1` | 24 features from four sensors: left and right shank and foot |
| Raw acceleration + DL | 85.0 ± 10.0 | 66.0 ± 18.5 | 60.2 ± 13.7 | 85.6 ± 7.3 | Accelerometer data. Binary classification (grouping: bad & mild). CNN-LSTM arch. 2 | All 12 sensors |

The input data was the acceleration, as the quaternions did not improve the results of acceleration in any of the cases. The results of options 1 and 2 were similar despite the variation in the temporal window. The F1-score obtained was $85.6 \pm 6.7$, and exceeded the ground truth, which was 66.7.

The difference in F1-score with the DTW + KNN model was 11.7, which was statistically significant ($p = 0.0272$). Although the distance with the RF model was 15.3, the large standard deviation of this model made the difference significant by a very small margin ($p = 0.0540$). Based on these results, we can confirm that the Deep Learning model was the best for classifying Test 3.

It was demonstrated by a range of training exercises that employ different sensor configurations that the most favorable outcome was achieved with a six-sensor configuration when executing Test 3. The positioning of these sensors should be as follows: pelvis, left and right hands, left and right feet, and head.

### Machine learning cluster analysis

Because the results were not too encouraging, with very high standard deviations, the hypothesis arisen to perform a *post hoc* analysis in which it was worth asking if the cause of the models not learning was not only because perhaps the ground truth used was not correct, but because there were actually other subgroups of participants, or evaluations, that were not being taken into account. In the first part of this analysis, a cluster analysis was performed to test whether participants can be grouped into more than two groups (cohorts). In the second part, a cluster analysis was performed to see if the evaluations of the activities can be grouped into more than two or three groups (scores).

#### Cluster analysis based on sociodemographic information and mini-BESTest scores

First, Fig. 5 shows the correlation matrix of the features. As can be seen, all activity tests of the 'Sensorial Orientation' category, 'Test 1', and 'Test 10' did not have variation, thus providing no information to the clustering algorithm, and the six features were discarded.

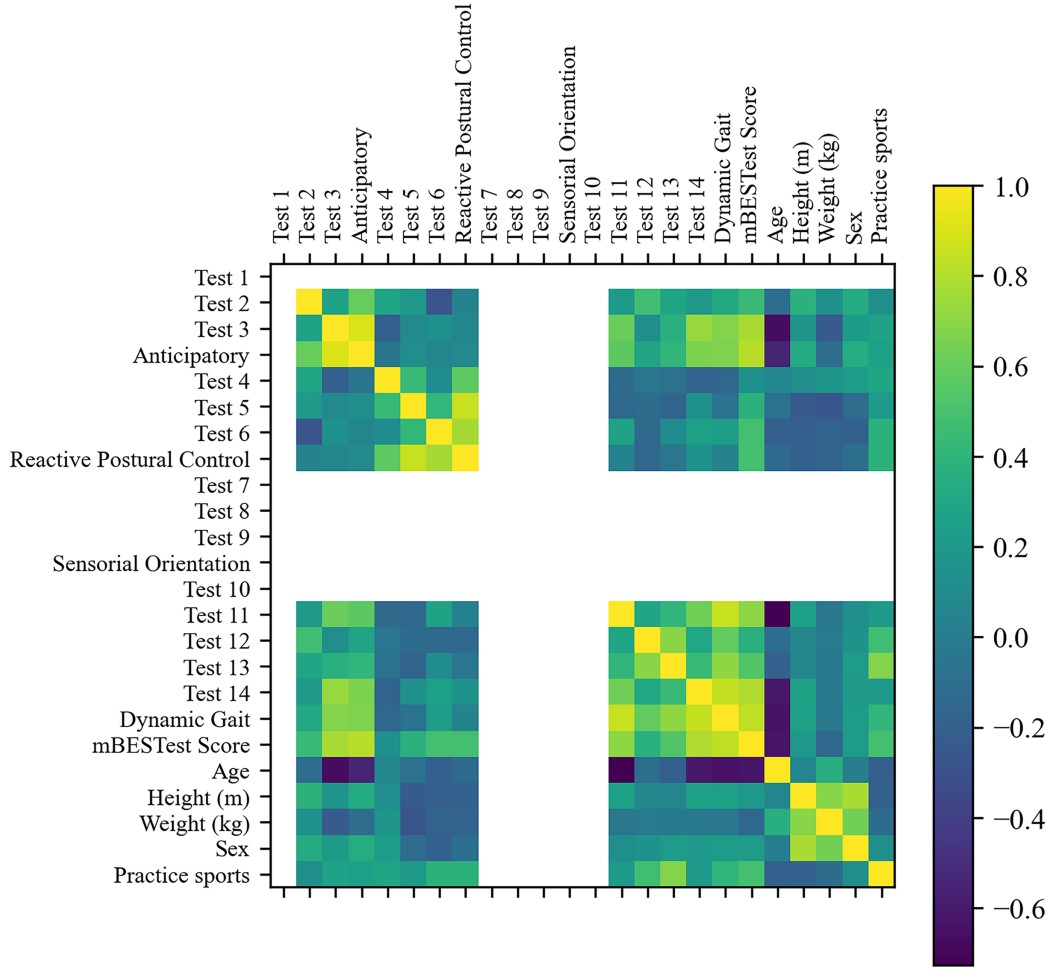

**Figure 5 Correlation matrix of sociodemographic information and mini-BESTest scores.**

Furthermore, by removing features with absolute correlation higher than 0.9, five more features were removed: 'Anticipatory', 'Test 4', 'Test 5', 'Test 6' and 'Reactive Postural Control'. Twelve features remained. The RFE selected the most relevant features for the cluster analysis. Despite the fact that 'Test 3' was the most relevant feature and a good clustering metric was achieved with only this feature, the best clustering metrics were achieved with three features: 'Test 3', 'Age' and 'Sex'.

Figure 6 shows the results of the cluster analysis for the best feature and the three best features after recursive feature elimination, respectively. The results showed that the optimal number of clusters was 2 for all metrics and set of features. It is worth mentioning that 'Test 3' is the most relevant feature and good clustering metrics are achieved only with this feature. However, the best clustering metrics are achieved with three features: 'Test 3', 'Age', and 'Sex'.

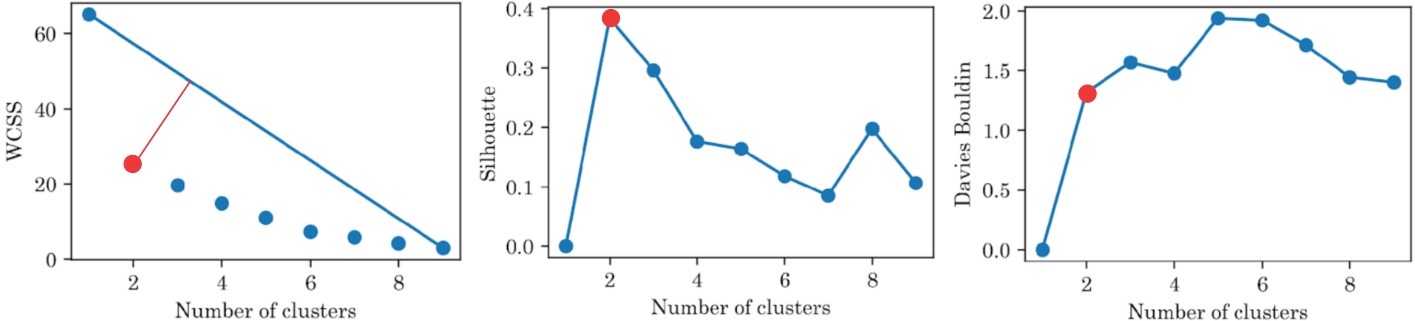

**Figure 6** Clustering metrics for sociodemographic information and mini-BESTest scores, for the best set of features: 'Test 3', 'Age', and 'Sex'. All metrics suggest that there are only two groups.

The clustering metrics were not limited and there was no level to compare with and, despite the fact that the clustering metrics were not high, Fig. 7 highlights that the participants were clearly separated into two groups, which corresponded to the original blind/sighted groups with a 90% F1-score (only one participant was misclassified).

Thus, the cluster analysis based on sociodemographic information and mini-BESTest scores suggested that there were two groups of participants: blind and sighted. An interesting aside result of this *post hoc* analysis was that a person can be classified as blind or sighted based only on the result of the 'Test 3', their 'Age', and 'Sex'.

### Cluster analysis based on IMU data

A cluster analysis based on IMU data was performed to determine whether the evaluations of the activities can be grouped into more than two or three groups (scores). The results of the cluster analysis are shown in Fig. 8. The best cluster analysis results were achieved with two features: 'torso_ay_mean' and 'torso_ay_energy'. The results showed that the optimal number of clusters was 2 for all the WCSS, Shilhouette, and Davies-Bouldin metrics.

These results indicated that time features based on acceleration signals were unable to group the evaluations of the activities into more than two scores, which was consistent with the results of the mini-BESTest evaluation.

It is worth mentioning that the 'duration' feature was then the next feature that appeared in the next best clustering, which was indicative that the duration of the activities was an important factor in the evaluation of the activities as expected. What is more interesting is that the energy of the torso acceleration signal was one of the most relevant features, indicating that the energy of the torso acceleration signal was an important factor in the evaluation of the activity 'Test 3'. However, this sensor was only selected in the DTW + KNN model and not in the RF model. This could be due to the fact that the torso was important, but its low variability was good for clustering, but not for classification. In fact, the subsequent features that appear are related to the foot and shank, which are the most relevant sensors for the classification of activity 'Test 3' in the RF model.

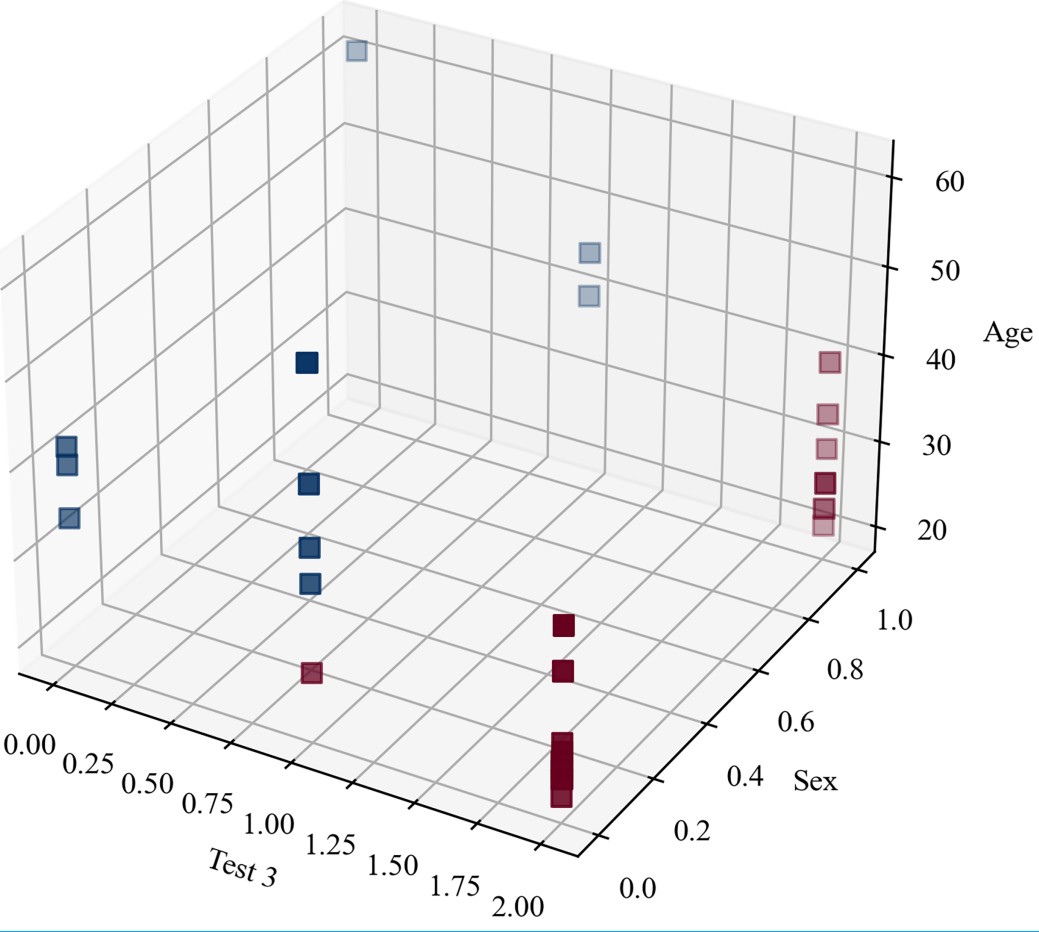

**Figure 7 Clustering blindness in mini-BESTest.**

## DISCUSSION

### Expert evaluation of mini-BESTest

In the present study, the mini-BESTest was evaluated by two experts: one a physiotherapist and the other a blind mobility teacher. Both experts observed video recordings of each activity. A similar process was employed by *Kamran et al. (2021)*, in which a physiotherapist ranked the activities that had previously been recorded on video.

### *Statistical analysis of mini-BESTest*

After the evaluation of all participants, a *post hoc* analysis was performed to find differences between sighed and blind people. As a result, the highest difference was found in Test 3, which evaluates the ability to maintain balance while standing on one leg. This result was also pointed out by *Rutkowska et al. (2015)* for visually impaired children, and also stated by *Blomqvist & Rehn (2007)*. In contrast, as indicated by *Kümmel et al. (2016)*, this generic one-leg stance test is specific enough to identify differences between blind and sighted individuals. However, as the authors suggest, each study needs its own specific test to evaluate balance, and this might be the case for the present study.

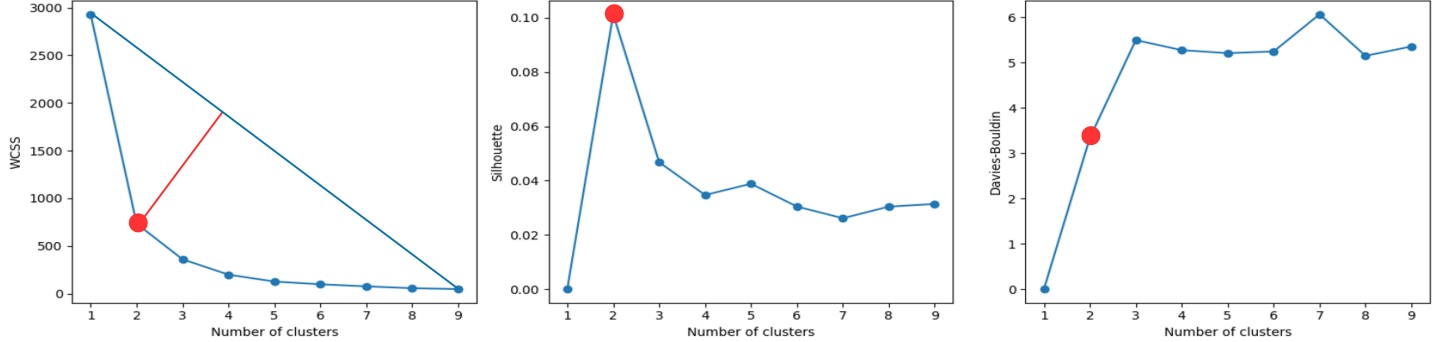

**Figure 8** Clustering metrics for IMU data, for the best set of features: 'torso_ay_mean', 'torso_ay_energy'. All metrics suggest that there are only two evaluation scores.                  

These findings emphasize that standing on one leg is a critical exercise for differentiating between blind and sighted individuals, as it directly measures static balance capacity. This insight supports the idea that focusing on this specific test could streamline the assessment process by reducing the need to perform the entire mini-BESTest. This approach aligns with the findings of *Pennell et al. (2024)*, who demonstrated that a reduced version of the mini-BESTest that focusses on key elements, including static balance tasks, is effective for assessing youth with visual impairments.

The analysis of signal duration revealed that the average duration of the exercise for blind individuals was lower than that of sighted individuals, and the agreement between expert evaluations and the re-evaluation by time was moderate. This discrepancy impacts the modeling of automatic classification of the score using supervised learning algorithms. Future work will address this factor to improve the reliability of automated assessments.

## AI-driven classification analysis

### Machine learning

Most research that includes balance tests and ML does not consider people with visual impairments and uses other balance tests. *Lin et al. (2022)*, predict the Berg Balance Scale (BBS) score of a participant without professional supervision using Machine Learning regression and IMUs. *Bao et al. (2019)* analyze the feasibility of using trunk sway data and ML techniques to automatically assess balance, providing accurate assessments at home.

### Deep learning

It has been confirmed that by employing DL models it is possible to treat IMU data as a time series, thereby enabling the classification of the punctuation given by a professional for a balance test. Thus, a binary classification has been determined as the most appropriate method to predict whether a participant has good or bad balance when performing Test 3 of the mini-BESTest. A similar procedure was followed by *Gil-Martín et al. (2021)* in founding the scores for the Y-Test, which was treated as a regression problem. This was in contrast to the desire to obtain the normalized reach distance (NRD) per each leg.

Regarding the number of sensors in the body for this experiment, information from 12 parts of the body was recorded and also analyzed to determine the minimum number of sensors to obtain acceptable performance in the classification of Test 3. The number of sensors was reduced from 12 to six sensors. In contrast, other studies included one, two or eight sensors depending on the activity. It could be because only with one sensor they could obtain one activity in their study, but ours included 14 different activities, and it was the first time a database was created with valuable information that takes into account blind people.

In addition, in this study four types of DL architectures were tested: CNN-1D, LSTM, GRU and CNN-LSTM being the hybrid model, the most accurate in classifying accuracy and F1-score. *Kim et al. (2021)* used a CNN-ID and GRU ensemble model to assess clinical balance using BBS. They compared it with previous work that used ML instead of DL and obtained 98.4% as the best accuracy of the model. Also, *El Marhraoui et al. (2023)* utilized a CNN to train and extract weights after training to predict fall events using the Balance Test Score.

Furthermore, in this work, data augmentation was used to improve classification performance and increase data for each class (mini-BESTest score), but the results did not exceed 60%, so it was discarded. As well as us *Wei, McElroy & Dey (2020)* used this technique to create more variable data from the Center of Mass (CoM) varying the position of x and z, and this with a Balance Evaluation Test to estimate the balance level of each subject if it has or does not fall risk.

Finally, when binary classification was tested using acceleration as input data, it was possible that labels 2 and 1 could be confused in the prediction due to a discrepancy in the evaluation process. To enhance the reliability of the results, it would be advisable to perform further tests with a larger number of data cases, including the evaluation of 0, 1, and 2 (to include a greater number of participants) and to retest the system. This approach could potentially lead to an improvement in system performance.

This study proposes an innovative methodology for classifying the mini-BESTest using IMU data, offering a promising approach to this field. However, it is imperative to acknowledge the limitations of the study. One key challenge is that only two professional evaluators participated in the initial classification, which can introduce bias into the metrics used. Including additional evaluators in future work could refine these metrics and subsequently improve the training and performance of the classification models.

Future work will focus on assessing the specific impact of exercises 3, 11, and 14, where significant differences in balance were observed for blind individuals. In addition, the development of new exercises and evaluation methods will be explored, emphasizing more accessible and widely practiced activities such as pilates, yoga, or taichi (*Wang et al., 2021*). These alternatives could provide complementary approaches to the mini-BESTest for assessing and improving balance in diverse populations.

## CONCLUSIONS

This study revealed notable discrepancies in the balance levels between blind and non-blind individuals, as reflected in the mini-BESTest score. The findings indicated that

standing on one leg corresponding to Test 3 is the most pronounced exercise, effectively differentiating between blind and sighted individuals.This item has to do with the person's static balancing capacity. This insight allows professionals to prioritize this exercise, which may reduce the need to perform the entire mini-BESTest.

In addition, our analysis of the acceleration data indicates a discrepancy between the evaluations made by the two experts and the established test criteria. However, AI models demonstrated potential, achieving an F1-score of 85.6% in binary classification contexts to predict scores evaluated by the mini-BESTest. This suggests that AI has the potential to provide a relatively accurate and objective measure of balance performance.

Furthermore, we investigated the possibility of alternative groupings of participants and scores using unsupervised clustering techniques. Cluster analysis demonstrated that IMU data alone were insufficient to effectively distinguish between the three levels of evaluation. However, the insights gained from the clustering can still inform the development of more nuanced and accurate classification models in the future.

Finally, the findings of this study can significantly help physiotherapists in conducting more objective evaluations of their patients' balance. Additionally, the results support the development of tele-rehabilitation systems tailored for blind individuals, enhancing the accessibility and effectiveness of rehabilitation programs.

## ACKNOWLEDGEMENTS

Milagros Jaén-Vargas would like to express her gratitude to ONCE, Madrid, Spain, particularly Fátima Peinado Villegas, for their guidance and support throughout the development process. Also, she acknowledges the active role that the organization played in recruiting volunteers for this study. In addition, she would like to thank Laura Brand and Carlos Arbós from the Blind Tenis group for helping to find volunteers. Finally, she would like to thank the blind and sighted volunteers who dedicated their time and effort to this study.

### Funding

Milagros Jaén-Vargas is funded by Instituto para la Formación y Aprovechamiento de Recursos Humanos and Secretaría Nacional de Ciencia, Tecnología e Innovación (IFARHU-SENACYT) grant (270-2018-968). Shiyang Li received scholarship funding from the China Scholarship Council (CSC) program grant (202308390098). The funders had no role in study design, data collection and analysis, decision to publish, or preparation of the manuscript.

### Grant Disclosures

The following grant information was disclosed by the authors:
Instituto para la Formación y Aprovechamiento de Recursos Humanos and Secretaría Nacional de Ciencia, Tecnología e Innovación (IFARHU-SENACYT): 270-2018-968.
China Scholarship Council (CSC): 202308390098.

## Competing Interests

The authors declare that they have no competing interests.

## Author Contributions

- Milagros Jaén-Vargas conceived and designed the experiments, performed the experiments, analyzed the data, performed the computation work, prepared figures and/or tables, authored or reviewed drafts of the article, and approved the final draft.
- Josué Pagán performed the experiments, analyzed the data, performed the computation work, prepared figures and/or tables, authored or reviewed drafts of the article, and approved the final draft.
- Shiyang Li performed the experiments, analyzed the data, authored or reviewed drafts of the article, and approved the final draft.
- María Fernanda Trujillo-Guerrero analyzed the data, authored or reviewed drafts of the article, and approved the final draft.
- Niloufar Kazemi analyzed the data, authored or reviewed drafts of the article, and approved the final draft.
- Alessio Sansò conceived and designed the experiments, analyzed the data, authored or reviewed drafts of the article, and approved the final draft.
- Benito Codina-Casals conceived and designed the experiments, analyzed the data, authored or reviewed drafts of the article, and approved the final draft.
- Roy Abi Zeid Daou analyzed the data, authored or reviewed drafts of the article, and approved the final draft.
- Jose Javier Serrano Olmedo conceived and designed the experiments, analyzed the data, authored or reviewed drafts of the article, leads the reaserching group, and approved the final draft.

## Ethics

The following information was supplied relating to ethical approvals (*i.e.*, approving body and any reference numbers):

The study objectives, experimental protocol, and informed consent were approved by the Ethics Committee of the Universidad Politécnica de Madrid (UPM) (ADPCEPCDVY-JJSO-DATOS-20230329).

## Data Availability

Data is available at Zenodo:

Jaén-Vargas, M., Pagán, J., Li, S., Kazemi, N., & Serrano-Olmedo, J. J. (2024). Automated balance assessment for blind and non-blind individuals using mini-BESTest and AI (v2.0.0) [Data set]. Zenodo. https://doi.org/10.5281/zenodo.13842814

Project code is available at GitHub: https://github.com/mjaenvargas/mini-BESTest_blind_noblind.

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
