# Peer review of "AI-driven balance evaluation: a comparative study between blind and non-blind individuals using the mini-BESTest"

_PeerJ Computer Science, doi:10.7717/peerj-cs.2695_

## Round 0.1 · original submission · Minor Revisions

Dear Authors,

Your paper has been reviewed. Based on the reviewers' reports, minor revisions are needed before it is considered for publication in PEERJ Computer Science. In particular, as the reviewers pointed out, the author must fix many typos and rewrite several statements to enhance their clarity in the updated version of their manuscript.

Reviewer 1 ·

Basic reporting

Balance and AI

This manuscript covers a very important topic and the authors should be applauded. There are some critical issues that must be revised before this manuscript can be published:

1. Pg. 2 lines 54-59 need a reference here.
2. Pg 2 lines 60-61 also
3. Pg 2 69-71 need reference
4. Pg 2 72-74 also
5. Pg 2 line 78 add this reference
Haibach-Beach, P., McNamara, S., & Lieberman, L.J. (2022). Home based
balance intervention of older adults with visual impairments. British Journal of
Visual Impairment, DOI: 10.1177/0264619620935937.

6. Pg. 2 94-96 need reference.
7. P3. 3 107-107-here you start sharing your purpose and you did not complete the introduction. You need to finish your “Why” before you state the purpose of the study.
8. Pg 3 line 137 again you start explaining your purpose/aim of the study. Move these to the end of the introduction.
9. Pg. 6 line 172-174-these are random percentages. How did you come up with these predictions? You need to be explicit.
10. Pg. 6 line 175-why do you use the word healthy? I see no need for this word here as it is assumed. Later in the article you continue to use the qword healthy for the sighted participants and not for the blind participants. This implies that they are not healthy. Just delete the word healthy throughout.
11. Your tense throughout the manuscript is present tense. You already completed this study so change the whole article to past tense.
12. Pg. 15 line 554-556 the sentence is confusing. Did the physiotherapist and the mobility teacher view the videos or just one of them? Please re-word this sentence to be clear.
13. Pg. 17 line 608 you can refer to the study by Pennell that reduces the items on this test for youth with VI
Pennell A., Fisher J., Patey M., Miedema S.T., Stodden D., Lieberman L. J., Webster C.,
Brian A. (2023). Measurement properties of Brief-BESTest scores from children, adolescents, and youth with visual impairments. Disability and Rehabilitation. Dec 1:1-10. doi: 10.1080/09638288.2023.2288935.

With some clarity this article can be a nice contribution to the existing literature

Experimental design

I am. not familiar with this statistical data at all

Validity of the findings

I assume they are valid and they make some sense although this is not my area of expertise.

Additional comments

Please see the comments in box 1.

Reviewer 2 ·

Basic reporting

I generally find this paper to be of a good standard. The investigation in use of Inertial Measurement Units (IMUs) and AI to assess balance in blind and sighted individuals is a useful study and appears to be making a novel contribution in an area in which the currently available literature is rather limited.
However there is space for improvement:
1. Experimental Setup and Data Acquisition: the descriptions of experimental setup and data acquisition process could be more detailed, e.g. the authors could provide more information about the specific conditions under which the data was collected (such as lighting, environment and the instructions given to participants).
2. Methodology on Modeling and Data Analysis: The overall workflow, which is given in Figure 2, is useful but some of these sections would benefit from clearer explanations and more elegant phrasing. For instance, the rationale behind the choice of specific hyperparameters for ML and deep learning models could be further elaborated.
3. AI methods: The authors could provide more context and reasoning for the choice of specific algorithms (i.e., DTW, KNN, Random Forest, and CNN+LSTM) and reasoning or detailing the choice of parameters.
4. Limitations of study: The authors could provide more thorough discussion of limitations of this study, giving detail and suggest directions for future research.

Experimental design

Overall, the experimental design is covered effectively. Although, descriptions of experimental setup and data acquisition process could be more detailed, e.g. the authors could provide more information about the specific conditions under which the data was collected (such as lighting, environment and the instructions given to participants).

Validity of the findings

The findings appear valid and well supported. I do not see further need for elaboration of this.

Additional comments

The paper is of good quality. The research question is well-defined and relevant, addressing a knowledge gap in the field of balance assessment for visually impaired individuals; as mentioned earlier there is currently rather limited literature on this subject. The methodology is sound, utilizing appropriate experimental design, data collection, and data analysis methods. The authors have done an adequate, brief and focused, literature review. The results are clearly presented and well-illustrated. The conclusions are suitable supported by the findings. The paper clearly adheres to ethical standards for which evidence of following the requisite procedures have been provided, and also provides appropriate acknowledgements and disclosures. Overall, I find that the paper demonstrates a strong understanding of the subject matter, following a rigorous research method, and provides a clear and useful contribution to the field. Therefore, I recommend that this paper be published.

---

## Round 0.2 · accepted · Accept

Dear Authors,

Based on the reviewer's report, your paper has been accepted for publication in PEERJ Computer Science. Thank you for your fine contribution.